# Kinetics of Guided Growth of Horizontal GaN Nanowires on Flat and Faceted Sapphire Surfaces

**DOI:** 10.3390/nano11030624

**Published:** 2021-03-03

**Authors:** Amnon Rothman, Jaroslav Maniš, Vladimir G. Dubrovskii, Tomáš Šikola, Jindřich Mach, Ernesto Joslevich

**Affiliations:** 1Department of Materials and Interfaces, Weizmann Institute of Science, Rehovot 76100, Israel; amnon.rothman@weizmann.ac.il; 2Institute of Physical Engineering, Brno University of Technology, Technická 2, 616 69 Brno, Czech Republic; Jaroslav.Manis@ceitec.vutbr.cz (J.M.); sikola@fme.vutbr.cz (T.Š.); mach@fme.vutbr.cz (J.M.); 3CEITEC (Central European Institute of Technology), Brno University of Technology, Purkyňova 123, 612 00 Brno, Czech Republic; 4Faculty of Physics, St. Petersburg State University, Universitetskaya Emb. 13B, 199034 St. Petersburg, Russia; dubrovskii@mail.ioffe.ru

**Keywords:** gallium nitride, nanowires, guided growth, surface-diffusion

## Abstract

The bottom-up assembly of nanowires facilitates the control of their dimensions, structure, orientation and physical properties. Surface-guided growth of planar nanowires has been shown to enable their assembly and alignment on substrates during growth, thus eliminating the need for additional post-growth processes. However, accurate control and understanding of the growth of the planar nanowires were achieved only recently, and only for ZnSe and ZnS nanowires. Here, we study the growth kinetics of surface-guided planar GaN nanowires on flat and faceted sapphire surfaces, based on the previous growth model. The data are fully consistent with the same model, presenting two limiting regimes—either the Gibbs–Thomson effect controlling the growth of the thinner nanowires or surface diffusion controlling the growth of thicker ones. The results are qualitatively compared with other semiconductors surface-guided planar nanowires materials, demonstrating the generality of the growth mechanism. The rational approach enabled by this general model provides better control of the nanowire (NW) dimensions and expands the range of materials systems and possible application of NW-based devices in nanotechnology.

## 1. Introduction

Semiconductor nanomaterials, in the form of nanotubes, nanowires (NWs) and nanoparticles, exhibit intrinsically unique physical properties due to size-confinement and geometric effects [1], which make them attractive building blocks for various applications in nanoscience and nanotechnology [2,3], NWs already play an essential role in optoelectronics [4,5], logic circuits [6,7] and quantum computing [8,9], owing to the possibility of controlling their size, structure, composition and morphology. One of the most common methods for fabrication of semiconductor NWs is the vapor–liquid–solid (VLS) growth process, where a metal catalyst on the substrate surface forms a liquid alloy droplet with the semiconductor material in the vapor phase and promotes the axial NW growth under it [10]. This process usually produces NWs with vertical or random out-of-plane orientations. Assembly of these nonplanar NWs into ordered planar arrays is one of the technical barriers toward their large-scale integration into practical devices. This can be achieved using several post-growth methods [11,12,13,14]. An alternative approach is the surface-guided growth of planar NWs, which is based on the assembly, alignment and positioning of the NWs in one step during their growth. Within this approach, the NWs’ growth is directed by the substrate, which can be achieved in different techniques including epitaxy, graphoepitaxy and artificial epitaxy [15,16,17,18].

As wide bandgap semiconductor nanomaterials [19], gallium nitride (GaN) NWs are of great interest because of its unique electronic properties, specifically for light-emitting diodes (LEDs) [20,21], lasers [22,23] and high-electron-mobility transistors (HEMTs) [24,25]. NW heterostructures based on GaN and its alloys present improved electrical and optical performance due to energy bandgap tuning [26,27,28,29]. Surface-guided GaN NWs were grown on different substrates and were guided by the three mentioned above modes, and were structurally, optically and electrically characterized [15,30,31]. However, fabrication of devices based on GaN and other semiconductor NWs is still challenging since the planar NWs have different lengths, and it might be difficult to have an optimal array with the required performance. Therefore, controlling the growth of the planar GaN NWs and understanding their growth mechanism is mandatory. Surprisingly, despite the large number of reports on GaN NWs, both nonplanar and planar, only a few studied the growth mechanism of GaN NWs. Maliakkal et al. [32] showed that for Ni-catalyzed GaN non-planar NWs, the Gibbs–Thomson (GT) effect controls the growth of thin NWs, while the diffusion-induced growth dominates for thicker NWs. The results were fitted by the common models established earlier for vertical NWs [33,34] which shows, in particular, the inverse correlation between the NW growth rate and its radius. Lim et al. [35] showed that the formation of vertical GaN NWs might be controlled by competition of a mono-nuclear versus poly-nuclear growth rather than the GT effect, where nucleation of NW monolayers occurs at the vapor–liquid–solid triple-phase line. Their model was successfully confirmed through experiments; however, it did not refer to any concrete mechanism of the material transport, such as direct impingement or surface diffusion of Ga atoms.

Despite in-depth research of different surface-guided semiconductor NWs over the last decade [36,37,38,39,40,41,42,43,44], the kinetics and mechanism of catalyst-assisted planar NW growth were studied comprehensively only recently and revealed the effect of dimensionality on the diffusion transport of the semiconductor materials into the droplet [45,46,47]. In particular, the model of Rothman et al. [47] fitted well the data on planar growth of ZnSe and ZnS NWs. The model of [47] considers different kinetic pathways of material transport into the planar NWs and shows that the planar growth is led by the two main effects—the GT effect for thinner NWs and the surface diffusion for thicker ones. The resulting expression for the NW growth rate dL/dt is given by [47]:(1)$$\frac{dL}{dt}=2\Omega I\left\{1-\theta_{lv}e^{R_{\mathrm{GT}}/R}+\left(1-\theta_{ls}e^{R_{\mathrm{GT}}/R}\right){\left(\frac{\lambda }{R}\right)}^{m}\right\},$$

Here, *I* is the vapor flux of the element which limits the growth, Ω is the elementary volume per pair in solid, $\theta_{lv}=n_{l}^{\infty }/\left(I\tau_{l}\right)$ and $\theta_{ls}=n_{l}^{\infty }/\left(I\tau_{s}\right)$ are the ratios of liquid (l)-to-vapor (v) and liquid-to-surface (s) activities, respectively,$\;\tau_{l}$ and $\tau_{s}$ are the characteristic lifetime of adatom in the droplet and the surface, respectively, $n_{l}^{\infty }$ is the activity of the infinite liquid phase (at R→∞), *R_GT_* is the characteristic GT radius, λ is the effective diffusion length of the adatoms, R is the NW radius. The power exponent *m* represents the dimensionality of the dominant diffusion pathway and can vary between 1 and 2, depending on the planar or non-planar configuration and the growth conditions. By fitting experimental data of the growth rates of differently sized NWs by Equation (1), one can deduce the GT radius *R_GT_*, the diffusion length λ and the power exponent *m*. The obtained value of *m* indicates the dominant diffusion pathway during the NW growth [47]. For *m* = 1, the main diffusion-induced contribution originated from atoms that impinge directly onto the NW side facets from the vapor and diffuse to the droplet, while the collection of the adatoms from the substrate surface is negligible. For *m* = 1.5 or *m* = 2, the adatoms are mainly collected from the substrate. The data for surface-guided ZnSe and ZnS surface-guided NWs grown on flat and faceted sapphire surfaces at different growth temperatures were well fitted to the model and revealed that planar NW growth was dominated by surface diffusion from the substrate with no significant effect of the surface morphology.

Here, we report the kinetics and growth mechanism of Ni-catalyzed surface-guided GaN NWs grown on flat and faceted sapphire surfaces based on the previous growth model. The results clearly show that the dimensionality of surface diffusion is close to 2, as predicted by the model for planar growth, and suggest that the main diffusion pathway is the adatom collection from the substrate. These results corroborate the generality of the proposed model and the critical role of the substrate within the guided growth approach.

## 2. Materials and Methods

### 2.1. NW Synthesis

Surface-guided GaN NWs were synthesized on R- and M-plane sapphire wafers (Roditi International, Inc., London, UK). The M-plane wafer was annealed at 1600 °C in air atmosphere using a high-temperature furnace (Nabertherm, Lilienthal, Germany), creating V-shape nanogrooves of the more thermodynamically stable S and R planes [48]. The catalyst pattern was defined using a standard photolithography process (MA/BA6 Karl-Suss mask aligner, Garching, Germany), a negative photoresist (NR-9 1000PY) and a suitable mask. Ni catalyst was deposited by electron-beam evaporation of a thin (~0.5 nm) film (Telemark, Battle Ground, WA, USA), followed by lift-off in acetone. Heating the substrate to 550 °C in air for 7 min was done before the synthesis to produce Ni nanoparticles with a variety of sizes to catalyze the subsequent VLS growth of GaN NWs.

The GaN NWs were grown by a chemical vapor deposition (CVD) method in a 3-zone tube furnace (Lindberg Blue M; Thermo Scientific, Waltham, MA, USA). The source materials were 0.3 gr Ga_2_O_3_ powder and 3.8 sccm of ammonia gas (99.999%, Maxima, Ashdod, Israel). The sapphire substrates were placed downstream on a quartz slide, and the tube was purged with a mixture of N_2_ and H_2_ 1300 and 90 sccm, respectively, at 400 mbar. The temperature of the source powder was held at 980–1000 °C, while the sapphire substrates were maintained at 950 °C. Typical growth time was 20 min.

### 2.2. NW Characterization

The synthesized GaN NWs were characterized using optical microscopy (Olympus BX-51, Tokyo, Japan) and field-emission SEM (LEO Supra 55 VP and Sigma 500 Zeiss, Oberkochen, Germany) over the working voltage range of 3–5 kV. The thicknesses of NWs, related to their radius, were measured using AFM (Veeco/Bruker, Multimode Nanoscope 7.30, Berlin, Germany) by applying the tapping mode and using 70 kHz Si-etched tips (Nanoprobes), while the lengths were measured using the SEM. The investigated NWs were examined throughout the entirety of their lengths to ensure they are not merged with others.

## 3. Results and Discussion

The study of surface-guided GaN NWs growth mechanism was carried out with Ni catalyst on flat and faceted sapphire substrates surfaces: the flat plane R$\left(1\bar{1}02\right)$ demonstrated the epitaxial guided growth while the faceted plane M $\left(1\bar{1}00\right)$ demonstrated the graphoepitaxail guided growth. The Ni catalyst was patterned as 3 × 30 µm rectangles by standard lithography including evaporation of a nominally 0.5-nanometer-thick Ni layer, followed by liftoff. The thin Ni films were thermally dewetted and converted to Ni nanoparticles before the growth. These nanoparticles were randomly dispersed within the pattern edges. The planar GaN NWs grow on flat R-plane sapphire along the $\left[\bar{1}101\right]$ direction of the sapphire substrate, presenting wurtzite crystal structure with an axial growth direction of $\left[10\bar{1}0\right]$ (Figure 1a–c). Planar GaN NWs on the faceted annealed M-plane sapphire substrate grew along the $\left[1\bar{2}10\right]$ direction of the substrate, with the same wurtzite crystal structure, presenting graphoepitaxial growth along the nanogrooves with an axial growth direction of $\left[14\bar{5}3\right]$ (Figure 1d–f). The NW densities are similar for both substrates (M-plane and R-plane sapphire). Our previous work provides comprehensive details about the crystal phases, epitaxial relations, and crystallographic orientations for similar GaN planar NWs grown under the same conditions [15]. We checked that the GaN NWs in the present work were similar to those previously characterized. Therefore, these characterizations are not presented here. The NWs have a Ni droplet at their ends, as seen in Figure 1c,f, which is a characteristic of the VLS tip-growth mechanism. A clear morphological difference between the GaN NWs on annealed M- and R-sapphire planes is in their thickness, as seen in the atomic force micrsocope scans shown in Figure 2. GaN NWs grown on the faceted M-plane are thinner (10–25 nm) than those on the flat R-plane (20–50 nm), as shown in Figure 3a,c. The reason for that difference should be attributed to the fact that for the faceted surface the NWs are confined by the annogrooves with a width of ~30 nm, which do not exist in the flat surface; hence, the NWs can grow thicker on the flat R-plane.

The growth rates of the planar GaN NWs were calculated by dividing the NW length by the growth time in the first approximation. The investigated NWs were examined throughout the entirety of their lengths to ensure they were not merged with others. The growth rates versus their radius of about 50 NWs are presented in Figure 3a,c for GaN NWs on annealed M-plane and R-plane sapphire substrates, respectively. The data present no clear correlation between the growth rate and the NW radius due to a large scatter. The large variance of the NW lengths can be attributed to different incubation times of individual NW, as discussed in depth in our previous work [47].

One way to overcome the large scatter of the data is to consider only the envelope of the upper points that represents the maximum NW length for a given radius and corresponds to NWs which have emerged earlier than the others. Such analysis reveals a clear trend: increasing of the growth rate with the radius of the thinnest NWs, reaching the maximum growth rate at an optimal radius corresponding to the longest NWs, followed by a long decreasing tail, as shown in Figure 3b,d. Fitting the envelope data by Equation (1) is presented in the same figures, with the parameters summarized in Table 1. The growth rates for both GaN NW samples does not start from 0, meaning that the Ni-catalyst droplets are larger than the minimum size dictated by the GT effect [34]. The fitted dimensionalities of surface diffusion are *m* = 2.03 ± 0.2 for GaN NW growth on the faceted surface and *m* = 1.93 ± 0.2 for growth on the flat surface. The values are close to 2, as predicted by the model, and suggest that the main diffusion pathway is the collection of Ga adatoms from the substrate, with the Ga diffusion lengths on the substrate and the NW sidewalls being on the same order of magnitude.

The deduced values of the *m* for planar GaN NW appears very similar to those reported for ZnS planar NWs grown on flat and faceted sapphire substrates [47] (*m* = 1.9 ± 0.2 and *m* = 1.8 ± 0.1, respectively). This comparison suggests that planar growth of GaN NWs follows the same diffusion pathway as for ZnSe and ZnS NWs. Furthermore, we may conclude that surface-guided growth of planar NWs is general and is dominated by two-dimensional surface diffusion. This mechanism is different from the one controlling the growth of non-planar NWs, which was shown to be predominantly one-dimensional surface diffusion [34,47]. This important difference can be explained as follows. In non-planar VLS NW growth, adatoms need to diffuse along the NW sidewalls to reach the catalyst droplet at the tip. This process is one-dimensional, which is why the dimensionality of the dominant diffusion pathway is close to *m* = 1. In planar VLS NW growth, adatoms do not need to climb along the NW sidewalls to reach the droplet as they are able to arrive directly from the two-dimensional substrate. Therefore, the dimensionality of the dominant diffusion pathway is close to *m* = 2.

The deduced values of the diffusion lengths λ for NWs grown on the flat and faceted surfaces are 41 and 47 nm, respectively. The collection area from which Ga adatoms diffuse toward the sidewalls of each individual NW is far from all the neighboring NWs, and hence the NW growth kinetics is not affected by the NW spacing. The maximum NWs growth rate is ~1.3 μm/min for both NWs grown on the flat and faceted surfaces. The optimal radius corresponding to this maximum growth rate is around 15 and 30 nm for the growth of planar GaN NWs on annealed M-plane and R-plane sapphire, respectively. The nominal growth rates *ΩI* deduced from the fits appear different for the two types of substrates—it equals only 0.63 µm/min for the faceted M-plane substrate and is much higher (5.67 µm/min) on the flat R-plane substrate. This large difference can be easily understood through the different temperatures of the source material, which is lower when the NWs are grown on the annealed M-plane. The characteristic GT radii *R_GT_* are very close in both cases, around 5–6 nm, which seems reasonable because this parameter is determined by the surface energy of the droplet [47]. This cut-off radius can also be seen in Figure 2b. Short NWs with catalyst size of ~5 nm can be observed in the center of the AFM scan. These NWs could appear from Ni residue particles during the lift-off outside the lithography-defined Ni.

## 4. Conclusions

In this article, the growth kinetics of Ni-catalyzed planar GaN NWs on flat and faceted sapphire substrates is investigated and quantified using a model, which considers the GT effect and surface diffusion of adatoms as the dominant effects influencing the NW growth rate. The NWs growth rate presents a low power dependence on the inverse NW radius. The power exponent *m* represents the dimensionality of the dominant diffusion pathway and equals 2.03 for growth on the faceted surface and 1.93 for growth on the flat surface. This indicates two-dimensional diffusion of Ga adatoms as the main pathway for material transport. The fact that the VLS growth of planar GaN NWs is governed by the same diffusion pathway for different substrates morphologies, and that very similar results were previously obtained for planar NWs of other semiconductor materials, strongly supports the generality of the growth mechanism of planar NWs. Competition between the GT effect and surface diffusion yields an optimum NW radius corresponding to the maximum length under a given set of growth conditions, which enables one to obtain the longest NWs with a better size uniformity.

These conclusions are remarkably similar to those drawn from our studies on the kinetics and mechanism of planar ZnS and ZnSe NW growth, recently reported, Ref. [47] despite notable differences between these two types of materials and their growth processes: ZnS and ZnSe NW grew from Au catalyst and metal chalcogenide vapors. Those chalcogenide NWs were prone to direct vapor–solid (VS) growth, which causes NW thickening in competition with the VLS elongation, and lead to relatively short NWs. In the present study, GaN NW grow from Ni catalysts and Ga vapor reacting with NH_3_, and elongate to much larger lengths without thickening, owing to the virtual absence of direct VS growth. The fact that such different processes follow similar kinetics and quantitatively fit the same theoretical model is very significant, regarding the generality and robustness of the proposed mechanism and theoretical model. This significantly raises the expectation that planar NWs of other materials will grow by a similar mechanism following this kinetics where two-dimensional surface diffusion plays a critical role. Specifically, GaN NWs are important nanomaterials ZnS and ZnSe NW and hence understanding the kinetics of their planar growth has an intrinsic value independently of analogous studies previously done on planar ZnS and ZnSe NWs.

This work shows that understanding the kinetics of the guided growth allows for better control over the dimensions of planar NWs and fine-tuning of their morphology. The obtained knowledge about the surface-guided GaN NWs can lead to regular arrays of NWs by using a single-size Ni nanoparticle catalyst that will enhance growth with the controlled and uniform lengths. This may help to extend the range of materials systems and possible applications of NW-based devices.

## Figures and Tables

**Figure 1 nanomaterials-11-00624-f001:**
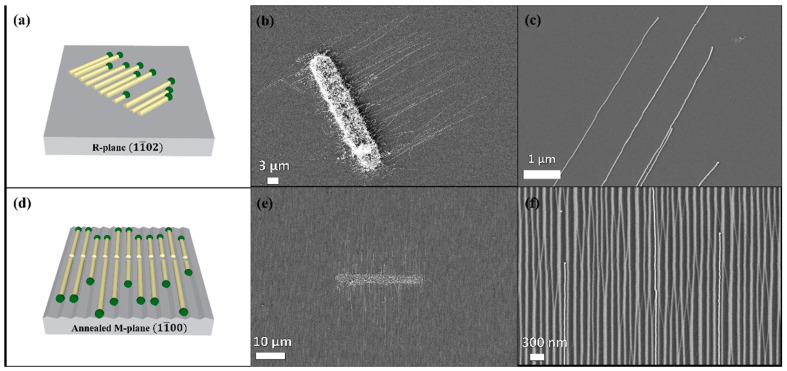
Guided VLS growth of surface-guided GaN nanowires (NWs). Schematic illustration of planar NWs on (**a**) R-plane and (**d**) annealed M-plane sapphire substrate. SEM images of planar GaN NWs grown on (**b**,**c**) flat R-plane sapphire substrate and on (**e**,**f**) faceted annealed M-plane sapphire substrate.

**Figure 2 nanomaterials-11-00624-f002:**
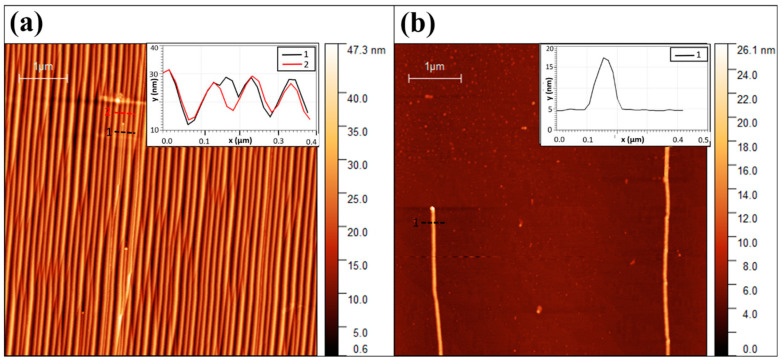
Atomic force microscopeimages of surface-guided GaN NWs grown on (**a**) the faceted annealed M-plane sapphire substrate and on (**b**) the flat R-plane sapphire substrate. Scale bar in both images is equal to 1 µm. The insets represent the height cross-sections of the NWs.

**Figure 3 nanomaterials-11-00624-f003:**
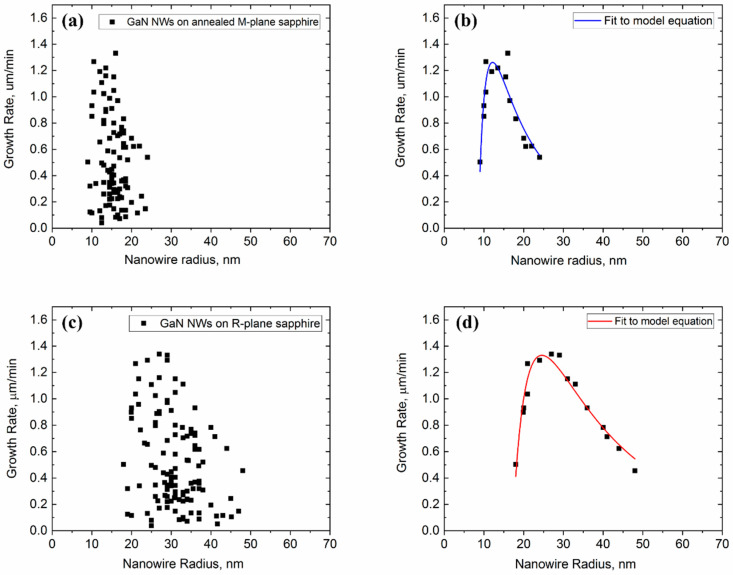
Growth rate versus radius of planar GaN NWs. (**a**) Data of NWs growth on faceted annealed M-plane sapphire substrate and (**b**) fitting of the envelope of the data by Equation (1). (**c**) Data of NWs growth on flat R-plane sapphire substrate and (**d**) fitting of the envelope of the data by Equation (1).

**Table 1 nanomaterials-11-00624-t001:** Fitting parameters of GaN NWs.

NWs	Sapphire Substrate	Source Temp., °C	Fitting Parameters					
			I, µm/min	θlv	θls	RGT, nm	λ, nm	m
GaN	AnnealedM-plane	980	0.63 ± 0.12	1.00 ± 0.05	0.47 ± 0.03	6.02 ± 0.13	40.93 ± 2.71	2.03 ± 0.15
GaN	R-plane	1000	5.67 ± 0.78	1.00 ± 0.05	0.69 ± 0.02	5.34 ± 0.09	46.85 ± 0.55	1.93 ± 0.17

## Data Availability

The data presented in this study are available on request from the corresponding author.

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
