# Peer review of "Kinetics of Guided Growth of Horizontal GaN Nanowires on Flat and Faceted Sapphire Surfaces"

_nanomaterials, 2021, doi:10.3390/nano11030624_

Round 1
Reviewer 1 Report
Dear Editor, dear Authors,
The manuscript presents theoretical and experimental work on surface-guided CVD growth of GaN nanowires with use of Ni as catalyst. The method is interesting and theoretical description is convincing, so I would recommend the manuscript for publication.
Comments:
1) Authors provided no information about quality of the NW: no X-ray analysis, no conductivity measurements neither luminescence. Judging from AFM images morphology is much worse than in the case of MBE grown NWs. Transition metals usually form fast recombination centers in III-V semiconductors, so GaN doped with Ni has probably low carrier lifetime and mobility and only weak luminescence.
2) In the line below Eq. (1), the symbols have bad format. More over some of them are not exxplained. Please explain meaning of: n_l^∞, τ_l, τ_s.
3) Lines 77 and 82:
Letters GT should be in subscript in symbol RGT, as in Eq. (1).
4) Lines 112-114:
Numbers should be in subscript in symbols Ga2O3, N2 and H2.
5) In chapter "Materials and Methods" (line 114) authors stated that source powder was held at 1000°C, while in table 1 there are two types of samples with source at 980°C and 1000°C.
6) NWs grown on M-plane have much higher density. What is the reason? Table 1 shows that flux was much higher for R-plane it seems to be in contradiction with so small number of NWs. The difference of source material temperature is really small 980°C vs 1000°C.
Author Response
Dear Editor,
Please find attached the revised version of our manuscript and our point-by-point response to the reviewers’ comments. Our answers (written in blue color) detail the changes made to the revised manuscript (written in red color) in response to each point. We believe that, after these revisions, the paper is improved, and you will find it suitable for publication.
Thank you very much for your consideration.
Sincerely,
Ernesto Joselevich.

Reviewer 2 Report
The authors investigated the growth mechanism of horizontal GaN nanowires and further compared to other semiconductor nanowires. It is a follow-up work compared to the paper published in Science in 2011. The “universal” model is interesting. The paper is well-written. One comment the authors might need to address is, as mentioned in the introduction, the authors claim that controlling uniformity (length) is mandatory, which is true for practical large area device applications. However, the results show in the present work still show uncontrolled uniformity. It would be better to comments on this and provide, e.g., approaches, to improve the uniformity.
Author Response

(The authors gave the same response as above.)
